# High-Spatial-Resolution Benchtop X-ray Fluorescence Imaging through Bragg-Diffraction-Based Focusing with Bent Mosaic Graphite Crystals: A Simulation Study

**DOI:** 10.3390/ijms25094733

**Published:** 2024-04-26

**Authors:** Kunal Kumar, Melanie Fachet, Christoph Hoeschen

**Affiliations:** Chair of Medical Systems Technology, Institute for Medical Technology, Faculty of Electrical Engineering and Information Technology, Otto von Guericke University Magdeburg, 39106 Magdeburg, Germany; kunal.kumar@ovgu.de (K.K.);

**Keywords:** X-ray fluorescence imaging, nanoparticles, HOPG/HAPG optics, Geant4

## Abstract

X-ray fluorescence imaging (XFI) can localize diagnostic or theranostic entities utilizing nanoparticle (NP)-based probes at high resolution in vivo, in vitro, and ex vivo. However, small-animal benchtop XFI systems demonstrating high spatial resolution (variable from sub-millimeter to millimeter range) in vivo are still limited to lighter elements (i.e., atomic number Z≤45). This study investigates the feasibility of focusing hard X-rays from solid-target tubes using ellipsoidal lens systems composed of mosaic graphite crystals with the aim of enabling high-resolution in vivo XFI applications with mid-Z (42≤Z≤64) elements. Monte Carlo simulations are performed to characterize the proposed focusing-optics concept and provide quantitative predictions of the XFI sensitivity, in silico tumor-bearing mice models loaded with palladium (Pd) and barium (Ba) NPs. Based on simulation results, the minimum detectable total mass of PdNPs per scan position is expected to be on the order of a few hundred nanograms under in vivo conform conditions. PdNP masses as low as 150 ng to 50 ng could be detectable with a resolution of 600 μm when imaging abdominal tumor lesions across a range of *low-dose* (0.8 μGy) to *high-dose* (8 μGy) exposure scenarios. The proposed focusing-optics concept presents a potential step toward realizing XFI with conventional X-ray tubes for high-resolution applications involving interesting NP formulations.

## 1. Introduction

Imaging techniques play a crucial role in present-day biomedical and fundamental research by elucidating complex biological processes through morphological, physiological, molecular, or functional information and thereby offering insights into the underlying pathology of diseases. A wide range of applications have benefited from the advent of in vivo molecular imaging methods that enable longitudinal studies at the tissue, organ, and systemic levels in intact vertebrate systems [1,2,3,4,5]. These applications range from pre-clinical screening of drug or therapeutic candidates, efficacy studies, and toxicology assessments to validation of therapeutic targets [1,3,4,5]. Furthermore, in vivo molecular imaging methods have also allowed researchers to gain a deeper understanding of disease biology and progression and to investigate biomarkers of disease prognosis and therapeutic response [6,7]. A variety of imaging techniques exist today for the above-said applications. These include positron emission tomography (PET), single-photon emission computed tomography (SPECT), bio-luminescence, optical fluorescence imaging, X-ray luminescence computed tomography (XLCT), magnetic resonance imaging (MRI), etc. [1,2,3,6,8].

While advances in micro-PET imaging have improved spatial resolution, there are intrinsic limits to achievable resolution driven by positron range, photon non-collinearity, and random events corresponding to the physical properties of Beta radiation [2]. Additionally, differentiation of multiple tracers is not plausible [2]. Micro-SPECT offers much better resolution (sub-millimeter) than micro-PET; however, in either case, the radiotracers decay over time, limiting the ability to track the same probes over extended periods [2,9]. Radiotracers used in PET/SPECT still present a high non-targeted radiation burden in mice under high-resolution requirements despite advancements in selective targeting and tracer clearance strategies, hampering longitudinal studies, especially those that involve radiosensitive biological systems [10,11,12]. Handling radiotracers and radiopharmaceuticals also presents additional challenges, restricting their flexibility. In methods that rely on detection or excitation with photons within the optical wavelength window, such as bio-luminescence and optical fluorescence imaging, the spatial resolution is constrained by depth-dependent optical scattering and attenuation to a large extent, typically reaching only millimeter range [8,13]. While methods relying on optical photon detection upon X-ray excitation, such as XLCT, offer improved spatial resolution, their reported radiation dose remains high, ranging from tens to hundreds of cGy [8,14].

X-ray fluorescence imaging (XFI) is emerging as a promising modality for biomedical molecular imaging. In this modality, X-ray fluorescence (XRF) photons are detected as signals that are emitted upon external excitation of tracer substances with X-rays. XFI offers a combination of high sensitivity and high spatial resolution at significant penetration depths within tissues and low exposure to ionizing radiation compared to present methods [9,10,13,15,16,17,18,19,20]. In contrast to PET/SPECT, XFI allows external control over radiation exposure, variable size scales (from variable resolution and beam positioning), and non-decaying substances, such as those based on nanoparticles (NPs). This enables serial, longitudinal, multiplexed, and targeted imaging at lower radiation doses [9,10,15,18,19,20,21]. Furthermore, XFI exhibits superior spatial resolution and depth of tissue penetration compared to current optical imaging methods while achieving resolution on par with computed tomography (CT) and MRI [9,13,15,17,18,19]. These aspects render XFI advantageous for various pre-clinical applications, such as long-term treatment response characterization, disease progression tracking, and applications requiring repeated measurements in the same subject (e.g., endogenous biodistribution studies, drug pharmacokinetics, contrast biodistribution studies, etc.) [9,10,17]. The potential for high-resolution XFI in human-sized objects has been shown in previous studies [16,20]. For a detailed review of recent developments and advancements in XFI, including scanning methods ranging from synchrotron-based pencil-beam to cone-beam benchtop X-ray fluorescence CT (XFCT), we refer readers to the following article [9].

While synchrotron-based small-animal XFI has demonstrated high performance in terms of spatial resolution, sensitivity, scanning time, and radiation dose, it comes at the cost of high infrastructural requirements [9]. Benchtop XFI systems, while more accessible, still face trade-offs between these performance factors [9,22,23,24,25]. However, recent studies using focusing optics with a liquid-metal-jet X-ray source have demonstrated promising results for achieving high in vivo spatial resolution, potentially bridging the gap between synchrotron and benchtop performance [18,19,21,26,27]. These studies combine the high brightness from a Galinstan (Ga/In/Sn alloy) target with focusing from a double-bounce multilayer Montel mirror that has been optimized for exploiting indium (In) $K_{\alpha }$ characteristic X-rays (∼24 keV). This delivers a quasi-monochromatic beam focused down to a resolution of around 100 μm. The spatial resolution in such an approach is, in principle, limited by the focused beam diameter, along with accompanying secondary scattering effects to a much lesser extent [18,19,28]. Despite its merits, currently used liquid targets still pose some limitations in XFI. Only those elements with K-edges below the In $K_{\alpha }$ emission line and sufficiently high fluorescence energies for adequate tissue transmission are usable as contrast agents [10]. This restriction currently applies to elements with atomic number Z≤45 and is also driven by the accompanying Compton background in addition to K-edge limitations [9,10,15,29,30].

For in vivo small-animal XFI applications with mid-Z contrast agents (elements with atomic numbers typically between 42≤Z≤64), we investigate the feasibility and prospects of focusing hard X-rays through ellipsoidally bent mosaic crystals. The tungsten $K_{\alpha }$ characteristic X-rays (∼59 keV) from a solid-anode microfocus tube are leveraged for this purpose. Our approach employs rotationally symmetric, hollow ellipsoidal substrates coated with multilayer graphite crystals on their inner concave surface. This geometry enables the capture of a larger X-ray beam cross-section (thereby, a larger solid angle) compared to, e.g., Kirkpatrick–Baez (KB) mirrors or side-by-side Montel mirrors [18,19,31]. Some of the well-known mosaic graphite optics (GrO) today include highly oriented pyrolytic graphite (HOPG) and highly annealed pyrolytic graphite (HAPG), which exhibit total integrated reflectivity an order of magnitude higher than perfect crystals due to their mosaic structure [32,33]. The present manufacturing process of HOPG/HAPG crystals allows the deposition of thin crystal layers on three-dimensional substrate structures, such as ellipsoids, while maintaining a highly oriented (low mosaic spread) arrangement [32,33]. This process presents a potentially cost-effective and straightforward method for realizing nested lens systems [32]. Implementing such nested ellipsoidal lens systems may prove beneficial for improving the reflection efficiency of X-rays in the hard regime, allowing higher flux density in the focused beams for high-energy applications [34]. Mosaic graphite crystals have been previously explored for XFI to defocus synchrotron beams for facilitating coarse scanning and for secondary filtering in benchtop systems with radial leaf collimators [35,36]. Their utility as Bragg-reflecting curved monochromators for focusing X-rays has also been explored in other techniques, such as X-ray spectrometry [33,37,38,39,40,41,42,43,44]. For further details regarding the production procedure and properties of HOPG/HAPG crystals, we refer readers to the following articles [32,42].

In this work, we implemented only a single-ellipsoid lens system for X-ray focusing. The optical properties of the proposed focusing concept are characterized, and the corresponding sensitivity with realistic NP concentrations for small-animal XFI is analyzed in silico using Monte Carlo (MC) simulations. Several well-validated MC codes, including FULKA, EGSnrc, PENELOPE, Geant4, and dedicated GPU-based models, have been used in the past for simulating XFI and XFCT for applications ranging from table-top to synchrotron-based setups [16,17,28,45,46,47,48]. In this work, the Geant4 toolkit is used, which is an open-source software package to simulate the passage of particles through matter from particle generation and propagation to physical interaction using relevant physical models and processes [49,50,51]. The utility of palladium (Pd) and barium (Ba) as contrast agents is explored in our work owing to their relatively weaker energy-dependent attenuation compared to low-Z elements in small-animal-sized objects (e.g., mice). Previous studies have investigated the suitability of Pd as a contrast agent for multi-tracking (multiplexing) macrophages and T-cells in synchrotron-based XFI of mice [15,17]. PdNPs have also been explored in vivo using tumor-bearing mice models for various other cancer imaging and therapy applications, including photoacoustic/CT imaging, drug delivery and activation, and combined photothermal–chemotherapy [52,53,54,55]. While Ba is presently utilized clinically as microscale ${\mathrm{BaSO}}_{4}$ particles, ${\mathrm{BaSO}}_{4}$ NPs are garnering interest as potential nanocarriers and contrast agents, and are currently under investigation [56,57,58]. In our work, simulations are performed on digital mouse models bearing tumors and loaded with either multiplexed PdNPs and BaNPs or singleplex BaNPs [59,60]. Two X-ray detector types are evaluated, i.e., silicon drift detector (SDD) and cadmium telluride (CdTe). Challenges associated with focusing hard X-rays for in vivo XFI applications using conventional sources are underscored, and potential avenues to improve focusing efficiency, sensitivity, and spatial resolution are outlined.

## 2. Results and Discussion

### 2.1. X-ray Focusing for Small-Animal XFI Applications Using Bent Mosaic Crystals

Firstly, MC simulations were performed to characterize and quantify the X-ray beam focusing and reflectivity properties of ellipsoidal and additionally planar GrO. These simulations investigated multilayer Bragg-reflecting crystals with crystallite orientations having mosaic spreads of m=0.12°, 0.24°, and 0.46° and corresponding crystal thicknesses of 80μm, 100μm, and 120μm, respectively.

Figure 1a shows the relationship between the transversal FWHM (full width at half maximum) of the focused X-ray beam in air, estimated using projection and cross-sectional line profiles of the photon intensity distribution at the focal point f=320 mm and the FWHM of the X-ray source spot. A corresponding normalized photon intensity distribution is shown in Figure 1b for m=0.12°.

As seen in Figure 1a, the size of the focal point exhibited a relatively linear relation to the size of the source spot. Increasing mosaic spread (and, consequently, the crystal thickness) led to more-pronounced broadening of the focused beam. This suggests a cumulative broadening effect stemming from an extended angular distribution of photons on the concave GrO surface or increased angular acceptance when the size of the source and mosaic spread of crystallites are increased. A lower limit on the transversal FWHM, likely attributed to the dominating mosaic effect, was observed for source sizes below approximately 60μm. Beyond this limit, the transversal FWHM of the focused beam plateaus, indicating no further enhancements in the so-called “beam resolution”. A source size of 320μm was chosen for XFI simulations in this work based on a reference X-ray tube (see Section 3.1.2); thereby, reaching a beam resolution of 600μm FWHM. Furthermore, photon step points were recorded in our simulations as they traversed each voxel plane to assess beam broadening within the implemented voxelized mouse model. The in-mouse transverse FWHMs of the beam at various longitudinal distances away from the focal point were observed to be well within the range of 600μm to 800μm.

To evaluate the performance of the Bragg-reflection simulation model, rocking curve analysis was performed on planar crystals. In Figure 2a, rocking curve profiles of planar GrO, evaluated with 59 keV monochromatic parallel beams, are shown. The narrowest rocking curve width, i.e., 0.3°, is seen for m=0.12° and has a peak reflectivity of approximately 9.5%. Correspondingly, Table 1 compares the GrO parameters between the reference data values taken from Grigorieva et al. [32], theoretical values derived from Equation (Equation 6), and those obtained from MC simulations. The data values cited from Grigorieva et al. [32] pertain to the Cu $K_{\alpha }$ emission energies, i.e., at around 8.03 keV. As seen in Table 1, the FWHM values of the rocking curve width obtained from MC simulations, $\delta^{\mathrm{MC}}$, at 8.03 keV are in good agreement with the reference values, δ [32], particularly for low mosaic spread (*m*). The peak reflectivity values $R_{\mathrm{peak}}$ calculated from MC simulations fall between theoretical and reference values. The deviation from theoretical values likely stems from multiple Bragg reflections not accounted for in the theoretical calculations. Under multiple reflections, the ray could also be reflected from the back face of crystallites and, consequently, travel out towards the crystal substrate. On the other hand, compared to the reference values from Grigorieva et al. [32], the results from our simulation did not account for certain real-world crystal defects (such as lattice defects) and experimental deviations that might be present in actual experiments. Moreover, unlike diffractometer experiments, the rocking curve analysis in our simulations did not include collimation of the primary or reflected beam; all contributions from Bragg reflections were recorded if they emerged from the crystal surface. This effect is evident as asymmetries in the rocking curve profile at rocking angles smaller than the Bragg angle $\theta_{B}$, as shown in Figure 2a for the 59 keV case (applicable to the 8.03 keV case as well).

Figure 2b compares the photon flux density (in units of photons per ${\mathrm{mm}}^{2}$) within a 1 ${\mathrm{mm}}^{2}$ area at the focal point between the direct X-rays and X-rays focused with the ellipsoidal GrO. The highest gain in flux density and reduction in the spectral bandwidth relative to the direct (unfocused) beam can be seen for m=0.12°. While a large mosaic spread (and, thereby, thickness) allows for higher integrated reflectivity, resulting in a higher intensity of reflected X-rays, it also introduces smearing at the focus. This smearing can be attributed to the parafocusing effect from deep-lying crystallites within the diffraction plane, which is further amplified by the larger mosaic spread and ultimately reduces the gain factors [61]. Additionally, to compare these results, ray-tracing simulations are carried out using the *mmpxrt* computer code (refer to Appendix A) [62,63]. Overall, key features of the spectral shape closely match; however, there are small differences, such as a broader spectral bandwidth away from the peak energies. Various factors could explain these differences. Firstly, in the ray-tracing simulations, only a 4 mm section of the ellipsoidal GrO curvature is utilized, as the current implementation of the *mmpxrt* software [62,63] imposes limitations on the crystal’s height above the base crystal surface lying on the xy-plane. Secondly, additional components of the focusing optics, including shielding elements and the X-ray beam-stop, are not implemented, as this cannot be done straightforwardly. Thirdly, these ray-tracing simulations do not consider other electromagnetic effects, such as Compton scattering or additional fluorescence from optical components.

Table 2 shows the primary performance parameters of the ellipsoidal GrO. As mentioned above, optics with the lowest mosaic spread (and, thereby, thickness) show the smallest transversal FWHM (about 600–800 μm) among the three simulated crystal types. The gain in intensity at peak energy $E_{peak}$ in reference to the direct X-ray beam within an area of 1 ${\mathrm{mm}}^{2}$ is also the highest in the lowest mosaic spread. Notably, this gain is approximately an order of magnitude for m=0.12°, followed by six-fold and two-fold for m=0.24° and 0.46°, respectively.

In Table 2, the focused beam’s energy bandwidth $\Delta E_{5\%}$ is also shown and is given in full width at 5% of the maximum. From MC simulations, $\Delta E_{5\%}$ at m=0.12° is smallest with a value of 5.8 keV, and the ratio of the energy bandwidth to the peak energy is $\Delta E_{5\%}/E_{peak}=9.7\%$. This corresponds to about a 65% reduction in spectral bandwidth relative to that of direct X-rays. The $\Delta E_{5\%}$ values of the focused beam in MC simulations suggest an approximately linear proportionality to the mosaic spread, as can be expected in a small-angle approximation for which the average incident photon angle and its spread on the crystal surface is $\theta_{inc}=1.78{}^{\circ}\pm 0.1{}^{\circ}$. The arrangement and geometry of the X-ray beam-stop and optical components contribute to this small spread in $\theta_{inc}$, as mentioned in Section 3.1.3. The ray-tracing simulations also show a similar trend, where $\Delta E_{5\%}$ at m=0.12° is the smallest, with a value of 8.9 keV, and $\Delta E_{5\%}/E_{peak}=15\%$. However, ray-tracing simulations yield larger $\Delta E_{5\%}$ values compared to MC simulations due to the previously mentioned differences in their simulation components.

In XFI utilizing conventional (or microfocus) X-ray tubes, optimizing the balance between minimizing spectral broadening and maximizing flux density becomes of key importance for achieving the near-monochromatic X-rays necessary for high-sensitivity in vivo imaging. In this work, the predominant limiting factor in the context of in vivo imaging of mice is the total scanning time per session, which is primarily constrained by the effective duration of anesthesia [10,64]. From the point of XRF tomography using focusing optics, key scanning parameters that must be considered include the nominal continuous photon flux density per scan position, total scanning area, and the number of projections per tomography scan.

While advanced X-ray generation technologies involving liquid-jet targets offer high flux density by surpassing thermal loading issues inherent to solid targets, other factors like cost-effectiveness, availability, and compatibility with various heavy element contrast agents and subject sizes must also be considered [9,10,15,65]. The choice of a tungsten-anode tube in our work is based on its high-energy characteristic X-ray lines (suitable for XFI in relatively larger subjects), broad applicability across various areas, and established technology available readily for diverse applications [65,66]. Furthermore, in addition to XFI, conventional tubes still provide the flexibility to serve as a common source for other established and upcoming imaging modalities, such as multi-contrast imaging with phase and dark-field contrast, spectral CT imaging, micro-CT, cone-beam CT, and dual-energy CT, among others [65,66]. Despite prevailing challenges associated with present-day X-ray sources and the need for compelling alternative solutions, conventional (or microfocus) tubes still serve as the workhorse in applications spanning clinical, pre-clinical, and research fields [65]. The proposed optics in this work could have potential applicability with both conventional and microfocus tubes contingent on the size scale of the subject being scanned and the required beam resolution.

With the proposed optics setup, 300 W (320 μm spot size) source target power, and 0.4 mm hafnium (Hf) beam filter, a nominal continuous photon flux density of approximately $10^{8}$
$\mathrm{photons}\cdot {\mathrm{mm}}^{-2}\cdot s^{-1}$ in a 600 μm diameter (FWHM of the focused beam) could be achievable with the ellipsoidal GrO (m=0.12°). Under this consideration, taking 60 min as the in vivo scan time limit, a single XRF tomography scan consisting of 10 projections in a 360° scan, a 20 mm×6 mm scanning area per projection, and around $2.7\;\times \;10^{7}$ photons per step (*low-dose* mode) would yield 10 axial slices covering a 3D scan volume of 20 mm × 20 mm × 6 mm. A single axial slice, therefore, corresponds to around $2.7\times 10^{8}$ photons per step position (*high-dose* mode). This can be compared to a previous study [18] on molybdenum oxide NPs that utilized a liquid-metal-jet source with a multilayer Montel mirror for focusing. In that study, a local-region XRF tomography scan with a 40 mm×20 mm scanning area per projection yielded 30 projections within a scan time of around 60 min [9,18].

If, otherwise, a singular long-duration in vivo study is of interest, anesthesia protocols with monitoring exist, such as those using isoflurane, to achieve even up to 40 h survival of mice [67,68]. Herein, the extended time frame may permit a significantly higher number of photons per step along with an increase in the number of axial slices and projections acquired per tomography scan. On the other hand, if primary evaluation sites are known a priori, e.g., from imaging modalities like MRI or CT, further reduction in beam resolution could become feasible. In such scenarios, denser scans at much higher resolution could become possible for the same time limit; for instance, a beam resolution of 300 μm could be reached using a 150 μm source size and 150 W tube power [69]. This may potentially find applications where local attributes are desired, e.g., biodistribution of passively or actively targeted NPs on the organ level, tumor biology, tumor response and healthy tissue monitoring in radiotherapy research, multiparametric imaging of therapeutic–agent–target interactions, selectivity studies for drug screening, etc. [3,15,70]. However, this remains to be thoroughly investigated through experimentation. This focusing approach may also offer other advantages, such as variable resolution. By modulating the size of the source spot, the scanning beam could be easily adjusted to achieve either high-resolution in vivo imaging for fine details or a broader field of view depending on the size scale of the scanned biological subjects. While adhering to the 3Rs (replacement, reduction, and refinement) is essential in animal research, in vivo studies remain vital in scenarios for which comprehensive understanding is required [71,72,73]. For instance, in specific oncological investigations, assessing the combined tumoral and immunological response to nanomedicines and NP-based contrast agents necessitates in vivo studies [71,73]. It also becomes crucial when in vitro assays, which are predictive of in vivo responses, are either unavailable or require extended development timelines and factors such as co-morbidities or other complex biological parameters are challenging to assess [71,73].

Other means to enhance flux density could include utilizing smaller minor curvatures ($a_{0}<10$ mm) and reducing the proximity of optical components to the X-ray source by compactly arranging system components. The trade-offs and relative gain factors would, however, need careful evaluation. In the present work, we prioritized flexibility over this specific optimization. Yet other promising directions for improvement can be explored, including the utilization of concentrically configured or nested lens systems, which increase the collection solid-angle of the primary beam, and crystals with small mosaic spread that exhibit higher peak reflectivities at high energies, such as aluminum (Au), copper (Cu), or germanium (Ge) [34,74,75].

### 2.2. XFI Sensitivity Analysis: In Silico Mouse Model

XFI simulations were performed on a tumor-bearing digital mouse model, employing reference in vivo parameters typically considered in various mice studies [10,15,53,55,64,76,77]. Simulations were conducted using various tumor lesion targets to evaluate the proposed focusing optics concept for high-resolution XFI applications. These in silico evaluations aim to predict the detection sensitivity and assess its dependence on the spatial size of the targets.

#### 2.2.1. Palladium (${\mathrm{Pd-K}}_{\alpha }$) and Barium (${\mathrm{Ba-K}}_{\alpha }$) XFI at *low-dose* Mode with SDDs

XFI simulations with SDDs incorporated multiplexed Pd and Ba contrast agents at varying concentrations in high to low configurations. The following results correspond to the simulations at *low-dose* mode XFI with $2.7\times 10^{7}$ photons per step position. Six 1 mm thick silicon drift detectors (SDDs) are used (70 ${\mathrm{mm}}^{2}$ area—collimated to 40 ${\mathrm{mm}}^{2}$, XR-100FastSDD, Amptek Inc., Bedford, MA, USA [78]) and are placed at a 140° azimuthal angle to the incident beam direction.

Figure 3a shows a typical simulated X-ray spectrum from SDDs for the multiplexed contrasts. In this case, the energy range from 15 keV to 35 keV (which includes the ${\mathrm{Pd-K}}_{\alpha }$ signal region around 21 keV and ${\mathrm{Ba-K}}_{\alpha }$ signal region around 32 keV) displays a minimum of single and multiple Compton scattering events, making it a suitable energy region to optimize for sensitivity when utilizing contrast agents in the mid-Z range.

Figure 3b,c compare the changes in significance (Z, see Equation (Equation 9)) of ${\mathrm{Pd-K}}_{\alpha }$ and ${\mathrm{Ba-K}}_{\alpha }$, respectively, at different concentrations within tumor lesions of 0.5 mm, 1.25 mm, and 5 mm diameter across three organ regions: the subcutaneous tissue region (lower right flank), liver, and left kidney. These evaluations are carried out to assess the feasibility of XRF imaging with optics across superficial and deep-lying tumor lesions in mice. Two significance (Z) thresholds are used, i.e., Z=5 (corresponding to a false positive probability of $p=2.868\times 10^{-5}$%) and Z=3 (p=0.135%). The subcutaneous targets perform systematically better than liver and kidney targets across all concentrations above Z=3. Due to the implemented ventrodorsal irradiation geometry, liver targets exhibit higher Z than kidney targets. This arises because the fluorescence photons must traverse a substantially larger volume of background tissue before reaching the XRF detectors from the kidney targets compared to the liver targets. However, in an XFI tomography scan, this would be compensated for. For studies exclusively targeting subcutaneous regions, much higher Z values could potentially be achievable. In the ventrodorsal irradiation scenario, this would require the placement of all detectors on the same side of the mouse’s body and positioned orthogonally to the primary X-ray beam. Based on Figure 3b, the detection sensitivity (Z>3) for ${\mathrm{Pd-K}}_{\alpha }$ in the *low-dose* mode is estimated to be 0.66 wt%, 0.05 wt%, and <0.01 wt% (extrapolated to 0.0066 wt%) for 0.5 mm, 1.25 mm, and 5 mm subcutaneous targets, respectively. For liver targets, this limit corresponds to 0.66 wt%, 0.1 wt%, and 0.01 wt%, and for and kidney, they are >1 wt%, 0.33 wt% and 0.033 wt%.

The Z values of ${\mathrm{Ba-K}}_{\alpha }$, as seen in Figure 3c, are consistently lower than those of ${\mathrm{Pd-K}}_{\alpha }$ for all target sizes and organ regions. This outcome is to be expected, as the signal region of ${\mathrm{Ba-K}}_{\alpha }$ receives a more significant contribution from multiple scattered Compton photons. Correspondingly, the detection limit is estimated to be greater than 1 wt% at all three organ sites when considering 0.5 mm target spheres. For 1.25 mm and 5 mm targets in all three regions, the detection limit (Z=5) is 0.33 wt% and 0.033 wt%, respectively.

Figure 3d–f depict the Z values in relation to variations in tumor size and concentration across the three organ regions. The Z values are seen to generally scale with the amount of agent mass present within the primary beam volume, as also shown by Kahl et al. [17]; however, at higher concentrations in larger targets, Z deviates from this linear proportionality, likely due to self-absorption effects. Additionally, larger targets exhibit slightly better performance for the same amount of contrast mass than smaller ones. This can be attributed to a combination of factors, i.e., a reduction in surrounding background tissue volume relative to the increasing tumor volume and, to a lesser extent, the excitation of fluorescence photons by the peripheral parts of the focused beam and scattered events [17].

#### 2.2.2. Palladium (${\mathrm{Pd-K}}_{\alpha }$) and Barium (${\mathrm{Ba-K}}_{\alpha }$) XFI at
*high-dose* Mode with SDDs

XFI simulations were performed in the *high-dose* mode, which corresponded to a total of $2.7\times 10^{8}$ photons per step position. Again, six 1 mm thick SDDs were used in the same imaging configuration as that of the *low-dose* mode. Figure 4a shows the typical XRF spectrum obtained in a single scan position (here corresponding to the liver targets at the upper abdominal region of the mouse model).

Figure 4b,c depict the significance (Z) plots for ${\mathrm{Pd-K}}_{\alpha }$ and ${\mathrm{Ba-K}}_{\alpha }$, respectively, corresponding to 1 mm and 5 mm tumor targets in the three organ regions. For the 1 mm diameter targets with ${\mathrm{Pd-K}}_{\alpha }$ fluorescence, the detection limit (Z=3) is estimated at 0.05 wt% across all organ regions, whereas for 5 mm targets, this limit improves to 0.0033 wt%. On the other hand, for ${\mathrm{Ba-K}}_{\alpha }$ fluorescence, this limit corresponds to approximately 0.1 wt% and 0.0066 wt% for 1 mm and 5 mm targets, respectively. In all simulated scenarios, akin to the *low-dose* case, the Pd contrast demonstrated higher significance (Z) above the threshold than Ba for equivalent concentrations within the target spheres. When comparing within fluorescence lines, Figure 4a illustrates that the $K_{\alpha }$ fluorescence signal was much stronger than the $K_{\beta }$ fluorescence for either contrast agent, and there is a pronounced background existing in the ${\mathrm{Ba-K}}_{\beta }$ region.

#### 2.2.3. Barium (${\mathrm{Ba-K}}_{\alpha }$) XFI at
*low-dose* Mode with CdTe Detectors

The feasibility of using cadmium telluride (CdTe) detectors is also explored for small-animal XFI applications. The following results correspond to the simulations at *low-dose* mode with six 1 mm thick CdTe detectors (XR-100CdTe, Amptek Inc., Bedford, MA, USA [79]) placed at 150° azimuthal angle to the incident beam direction. Figure 5a shows a typical XRF spectrum with singleplex Ba contrast agents.

While CdTe detectors offer high detection efficiency at higher energies than silicon sensors, they exhibit significant escape peaks. These peaks arise due to incomplete photon energy depositions from the escape of correspondingly generated Cd and Te fluorescence photons. This creates additional background events with origins from multiple scattered Compton interactions centered around 25 keV, leaving only a limited valley-like region between roughly 30 keV to 37 keV for any optimization.

Figure 5b shows the significance (Z) for ${\mathrm{Ba-K}}_{\alpha }$ fluorescence in the *low-dose* mode. In this case, the detection limit (Z=3) corresponds to Ba concentrations of >1 wt%, 0.33 wt%, and 0.033 wt% for 0.5 mm, 1.25 mm, and 5 mm diameter tumor targets, respectively.

#### 2.2.4. Barium (${\mathrm{Ba-K}}_{\alpha }$) XFI at
*high-dose* Mode with CdTe Detectors

Figure 6a,b show the typical spectrum and significance (Z) for ${\mathrm{Ba-K}}_{\alpha }$ fluorescence in the *high-dose* mode. Simulations in the *high-dose* mode show improved significance (Z) for ${\mathrm{Ba-K}}_{\alpha }$ fluorescence, with values exceeding the Z=5 threshold for both 1 mm and 5 mm diameter targets. Consequently, targets within all three organ regions demonstrated detection limits below 0.33 wt% and 0.033 wt% for 1 mm and 5 mm targets, respectively.

To contextualize the estimated detectable mass attainable through the proposed focusing optics, we reference previously reported values from studies on the mass accumulation of palladium-based contrast agents in tumor sites and other regions [15,17,53,54,55].

Using synchrotron XFI, Staufer et al. [15] reported total reconstructed PdNP masses as low as 171.1 ng around the mouse lumbar region. Assuming that the minimum pixel values in their XFI maps represent the threshold of detection, then this value translates to approximately 20 ng PdNP per ${\mathrm{mm}}^{2}$ [15]. Nonetheless, based on simulation results by Kahl et al. [17], detection limits of <3.3 μg/mL and 10 μg/mL were reported for 5.5 mm diameter subcutaneous and central targets, respectively. In the case of subcutaneous targets, this amounts to around 0.29 μg Pd mass within the tumor volume and approximately 18 ng Pd mass within the primary beam volume, assuming a cylindrical intersection with the tumor volume. Similarly, for central targets, these limits translate to around 0.9 μg Pd mass within the tumor volume and approximately 55 ng Pd mass within the primary beam volume.

Miller et al. [53] investigated the efficiency and safety of Pd-NP (nano-palladium) at a maximum-tolerated dose (MTD) of 50 mg/kg in two distinct xenograft mouse models: models with subcutaneous human fibrosarcoma tumors and orthotopic ovarian cancer (OVCA) models using the human ES2 cell line. Using inductively coupled plasma mass spectrometry (ICP-MS), Pd accumulation in the bulk tumor mass of around 0.5±0.1% injected dose per gram of tissue (%ID/g) after 24 h is reported in OVCA models [53]. Using the tumor growth curves from this reference, the mass accumulation of Pd-NP in 24 h thereby amounts to approximately 80 ng assuming a homogeneous sphere of 5.8 mm diameter (100 ${\mathrm{mm}}^{3}$ volume). For the fibrosarcoma model, Pd accumulation of 0.32±0.04 %ID/g after 24 h is reported [53]. This amounts to approximately 53 ng in the 5.8 mm diameter sphere. Moreover, for all models, a range of 5–20 μM Pd-NP concentration is reported at 50 mg/kg MTD, amounting to around 53–213 ng Pd-NP mass in the assumed sphere [53].

Shi et al. [54] studied the in vivo biodistribution of Pd@Au–PEG-Pt/Au–PEG-Pt in S180 tumor-bearing mice models, while Chen et al. [55] investigated PEGylated Pd@Au nanoplates (Pd@Au-PEG) in 4T1 murine breast cancer models. Shi et al. [54] reported a high tumor accumulation of Pd@Au–PEG-Pt and Au–PEG-Pt of 29 %ID/g and 10 %ID/g, respectively, as measured by ICP-MS, thereby translating into mass accumulation of 1.83 μg and 3.15 μg in the tumor. Chen et al. [55] previously found significantly higher tumor accumulation of Pd@Au-PEG, reaching 2 %ID/g, 15 %ID/g, and 79 %ID/g at 1 h, 12 h, and 24 h post-injection, respectively. This corresponds to approximately 0.6 μg, 4.7 μg, and 24.9 μg Pd@Au-PEG mass in a 5.8 mm diameter spherical tumor volume.

Taking an example case from our simulations, a detection limit of 0.01 wt% (Z=5) is estimated for 5 mm subcutaneous and liver targets in *low-dose* mode XFI, corresponding to approximately 100 μgPd per gram of tumor weight (100 μg/g). This translates into a minimum detectable total Pd mass of roughly 7 μg within a spherical tumor volume, assuming an average diameter of 5 mm. As mentioned earlier, the subcutaneous targets outperform liver targets; however, the margin is smaller than would be expected. The ventrodorsal irradiation geometry limits the number of detectors that can achieve an optimal view. Only three detectors can directly detect the XRF photons without them traversing significant background tissue. The remaining three detectors receive the rays that must traverse more than an average torso thickness, specifically from the right flank to the left lateral region, thereby attenuating the XRF signals. For the *high-dose* mode XFI, this detection limit improves to roughly 0.0033 wt% (Z=5), corresponding to 33 μgPd/g tumor weight and translating into a minimum detectable total Pd mass of about 2.2 μg within the same tumor volume. However, it is noteworthy that the actual beam cross-section interacting with the spherical tumor target is much smaller, i.e., 600 μm FWHM, compared to that of the 5 mm diameter. Considering this, we can approximate a cylindrical intersection volume between the primary X-ray beam and the tumor volume [17]. In this case, the minimum detectable total Pd mass within the primary beam volume amounts to around 150 ng in the *low-dose* mode and 50 ng in the *high-dose* mode. Furthermore, assuming a 20 mm mouse torso thickness, the above consideration leads to an average concentration of approximately 26 μg/mL (or 242 μM) within the entire beam. Similarly, for the *high-dose* mode, this corresponds to an average concentration of about 8.5 μg/mL (or 80 μM) within the entire beam.

A configuration with only two SDDs operating in the *low-dose* mode is expected to achieve a detection limit (Z=5) above 0.033 wt% and 0.05 wt% for 5 mm subcutaneous and liver targets, respectively. This translates to a minimum detectable total Pd mass within the primary beam volume of approximately 500 ng and 750 ng, respectively.

Based on our simulations with X-ray focusing and solid-target tubes operating in the so-called *low-dose* mode, the detectable Pd mass is predicted to be at least three times higher than that of synchrotron-based XFI under in vivo conform conditions, only reaching comparable values in the *high-dose* mode [10,15,17]. Furthermore, our estimates shed light on where our predictions of Pd mass lie relative to the expected tumor accumulation masses of currently investigated Pd-based NPs and promising XFI imaging techniques [10,15,17,18,19,53,54,55].

Our simulations identify an optimal energy range of 15–35 keV for SDDs, coinciding with the region exhibiting minimum background contribution from single- and multiple-scattered Compton photons. These findings align with previous studies that have demonstrated similar optimal energy ranges for SDDs, specifically in the context of synchrotron-based XFI [10,15]. On the other hand, CdTe detectors would only be suited for XFI applications involving elements with much higher characteristic X-ray energies, such as gold nanoparticles (AuNPs) and larger imaging objects. While palladium exhibited the highest sensitivity in our simulations, the utility of Ba is also evaluated. Herein, Ba was chosen specifically due to its proximity to the upper limit of the determined optimal range. Nevertheless, based on our current findings, it can be identified that elements with emission energies within the 15–35 keV energy range could be usable. This may also include other promising candidates such as molybdenum (Mo), ruthenium (Ru), rhodium (Rh), silver (Ag), iodine (I), etc. [9,10,15,21]. Utilization of X-ray optics may offer promising progress towards using XFI with conventional X-ray tubes for various potential applications involving interesting NP formulations, e.g., within pre-clinical oncology, drug development, and more. However, further research and experimentation are warranted to evaluate the feasibility of the proposed XFI concept within the framework of pre-clinical or other applications.

### 2.3. Radiation Dose

The total dose deposited in organs and tissues per scan position is calculated for the voxelized mouse model based on energy depositions per hit within our simulations. Figure 7a,b compare these dose values between the *low-dose* and *high-dose* modes. These calculated doses represent the scenario where the tumor target is ideally positioned at the center of the scan system and beam cross-section, following a ventrodorsal irradiation geometry.

The tumor dose in a single scan position for either organ region, i.e., subcutaneous, liver, or kidney, is estimated to be on the order of 0.4 mGy and 3.4 mGy for the *low-dose* and *high-dose* modes, respectively. Central scan positions (i.e., around the liver) tend to deliver the highest full-body dose for a single position, followed by those scan positions that target the kidney and subcutaneous regions. The full-body dose for a single position in *low-dose* mode corresponds to about 0.8 μGy, 0.55 μGy, and 0.3 μGy for the liver, kidney, and subcutaneous regions, respectively. For *high-dose* mode, this amounts to an order of magnitude increase. A local scan of a 5 mm target in the upper abdomen region, assuming stepping to cover a planar scan area of 25 ${\mathrm{mm}}^{2}$, would require roughly 70 steps with a beam resolution of 600 μm FWHM, leading to an estimated full-body dose of 56 μGy and 0.6 mGy in the *low-dose* and *high-dose* modes, respectively.

An XRF tomography scan considering 10 projections, each covering a 20 mm × 6 mm scanning area per projection, would deliver an estimated full-body dose (i.e., for the entire tomography scan) of 2.7 mGy in the *low-dose* mode. Typical doses for longitudinal studies in mice have been reported to range from 17 mGy to 780 mGy for single micro-CT scans [64]. Damage repair over the course of several hours has also been reported in rodents at low radiation doses of around 300 mGy [64,76,77]. Thereby, considering a dose limit of 300 mGy, an entire mouse scan would result in roughly 215 projections (approximately 1.4 mGy per projection, each with a scan area of 20 mm × 85 mm) within the *low-dose* mode. Moreover, the full-body radiation doses in otherwise sparse or local tomography scans are expected to be well below typically reported values.

## 3. Materials and Methods

### 3.1. Monte Carlo (MC) Simulation and Setup

#### 3.1.1. Bragg Reflection Process in Geant4

The following section outlines the methodology employed to model and implement the Bragg reflection process in the Geant4 simulation toolkit (v. 10.05.p01) alongside standard electromagnetic (EM) models and processes.

A simulation model for the Bragg reflection process is presented in this work since, at present, X-ray diffraction processes have not been integrated into the publicly available version of the Geant4 software. For implementing the remaining low-energy EM processes, such as Compton scattering and photoelectric absorption, among other processes, the Penelope EM model is used and is implemented through the *G4EmPenelopePhysics* physics constructor [80,81]. In an approach similar to Guan et al. [82], two new C++ classes are developed: *BraggReflectionPhysics*, derived from Geant4 base class *G4VPhysicsConstructor*, and *BraggReflection*, derived from Geant4 base class *G4VDiscreteProcess*. The two physics constructors *G4EmPenelopePhysics* and *BraggReflectionPhysics* were added to a modular physics list derived from *G4VModularPhysicsList*. Subsequently, the Bragg reflection process was implemented as a discrete gamma (i.e., the X-ray photon in Geant4) process by adding *BraggReflection* to Geant4’s physical process manager, i.e., *G4ProcessManager*.

Within the new *BraggReflection* discrete process class, σ, a linear scattering coefficient is used to propose the step length for Gamma stepping within the crystal volume as explained below. For circular focusing configurations with crystals, X-ray diffraction may have Bragg reflection (same surfaces for the incident and diffracted X-rays) or Laue transmission (opposing incident and diffraction surfaces) geometries [34,82]. In this work, for X-ray focusing, we use mosaic graphite crystals arranged in an ellipsoidal focusing configuration, as described in Section 3.1.3. The defect structure of these crystals is such that several small scattering crystallite domains can be assumed, each approximating nearly perfect crystals, distributed in orientations described by a spread parameter, i.e., the mosaic spread (m). The reflecting power of a mosaic crystal layer of thickness dT is, therefore, calculated as [83,84,85,86]
(1)$$\sigma dT=W\left(\Delta ;m\right)R_{H}^{\theta_{B}}\frac{dT}{t_{0}},$$
where σ represents the linear scattering coefficient for diffraction, W(Δ;m) is the distribution function determining the crystallite orientation parameterized by *m*, Δ is the rocking angle, and $t_{0}$ denotes the crystallite thickness. The Bragg angle $\theta_{B}$ for X-ray wavelength λ, diffraction order *n*, and lattice spacing $d_{hkl}$ (=0.3354 nm for graphite crystals) is $\theta_{B}=\arcsin\left(n\lambda /2d_{hkl}\right)$. In this work, W(Δ;m) is taken as a Lorentzian distribution: (2)$$W\left(\Delta ;m\right)=\frac{1}{\frac{m\pi }{2}\left[1+{\left(\frac{2\Delta }{m}\right)}^{2}\right]}.$$

A Lorentzian-shaped mosaicity is used since crystals with low mosaicities are observed to follow a Lorentzian distribution rather than a Gaussian distribution [37,83]. In Equation (Equation 1), $R_{H}^{\theta_{B}}$ is the integrated reflecting power for a single scattering domain, given as [86]
(3)$$R_{H}^{\theta_{B}}=Q\frac{t_{0}}{\cos\theta_{0}},$$
where $\cos\theta_{0}$ is the direction cosine of the incident rays relative to the crystal normal, and *Q* is the average diffraction scattering cross-section per unit volume. This is calculated within the kinematical theory as [85,86]
(4)$$Q={\left|\frac{r_{e}F_{hkl}}{V}\right|}^{2}\frac{\lambda^{3}}{\sin2\theta_{B}}\frac{1+\cos^{2}2\theta_{B}}{2},$$
where $F_{hkl}$ is the structure factor for Miller indices (hkl=002 for this work), $r_{e}$ is the classical electron radius, and *V* is the unit cell volume. As a result of the production process, the crystallites of pyrolytic graphite exhibit a high degree of preferred orientation along the *c*-axis of the graphite structure, which is parallel to the surface normal but shows no preference for rotation around the *c*-axis [40,87]. The 00*l* reflections lead to well-defined diffraction spots with odd values forbidden (l=1,3,5…), while all other *hkl* diffractions result in homogeneous Debye–Scherrer rings [40,87]. In our simulations, the first-order reflection in the (002)-lattice plane, which exhibits the highest intrinsic diffraction intensity compared to other planes, is assumed as the predominant reflection mode considering the focusing configuration described in Section 3.1.3 and central energy corresponding to the tungsten $K_{\alpha }$ characteristic X-rays [39,40,41,43,44,88,89]. X-ray photons with energies central to the tungsten $K_{\alpha }$ emission ($E_{peak}\approx 59$ keV) are more likely to be reflected at the Bragg angle, $\theta_{B}\approx 1.79{}^{\circ}$, on the (002)-lattice. On the (004)-lattice plane, with the second-highest integrated reflectivity roughly 10% that of the 002-plane, X-rays are likely to be reflected with energies around twice the $K_{\alpha }$ energy (≈118 keV). Some of the key optical parameters employed for implementing the diffraction process in our simulations with geometries involving ellipsoidal and planar GrO are summarized in Table 3. The expected mosaic spread and thickness parameters were provided by Optigraph GmbH, Berlin, Germany, and the corresponding details can be found in Grigorieva et al. [32].

In the member function *PostStepGetPhysicalInteractionLength()* within the new *BraggReflection* class, the step length is calculated and proposed only if the post-step point of the Gamma photon is within the crystal volume. When inside the crystal volume, the Rayleigh scattering process is replaced by the diffraction process. Otherwise, if the post-step point is outside the crystal volume, this function returns the maximum finite representable floating-point number as the proposed step length. This ensures that the Bragg reflection process is limited to occurrences within the crystal volume alone. In this simulation model, multiple Bragg reflections can occur within the crystal bulk, along with other EM processes. Furthermore, to facilitate an accelerated simulation framework, we employ “virtual” crystallites, similar to Guan et al. [82]. However, in our case, orientations of the crystallites corresponding to each Cartesian axis are sampled from a Lorentzian distribution instead of a Gaussian, at each post-step point within the crystal volume during photon tracking. Finally, if the Bragg reflection process is initiated, the direction of the Bragg-reflected ray is proposed by the *PostStepDoIt()* function of the *BraggReflection* class. Here, the sampled crystallite normal and the incident Gamma direction are used to determine the reflected ray vector.

#### 3.1.2. X-ray Source Model

A reflection-type static target bremsstrahlung tube is modeled in Geant4 to generate hard X-rays intended for the focused-beam XFI. For this purpose, we use a reference microfocus X-ray tube (XWT-225-XC, X-RAY WorX GmbH, Garbsen, Germany) with a tungsten target, a nominal tube voltage of 225 kV, a nominal anode input power of 300 W, and a maximum beam current of 3 mA [69]. Tungsten exhibits a very high melting point and high thermal conductivity compared to other available materials, which is necessary for effective heat dissipation, which, in turn, allows high loading capacity for high-flux applications [90]. The conversion to bremsstrahlung energies from accelerated electrons exhibits a rather high efficiency for high-atomic-number elements [90]. In this work, the $K_{\alpha }$ characteristic emission lines of the tungsten (i.e., $K_{\alpha 1}=59.32$ keV and $K_{\alpha 2}=57.98$ keV) tube are exploited for beam focusing with ellipsoidal GrO. Due to the higher energy of tungsten’s characteristic lines compared to the K-edge of most mid-Z contrast elements, XRF excitation of these elements is facilitated, avoiding interference with their K-shell XRF signals. An excitation voltage of 150 kV is used to allow high-intensity emission of tungsten $K_{\alpha }$ X-rays while maintaining sufficient beam current. For 150 kVp and a target power of 300 W, a stable source focal spot size of 320 μm is attainable [69]. The primary beam is pre-filtered with a 0.4 mm hafnium (Hf) sheet and collimated before reaching the optics using a lead aperture. Here, the Hf K-edge absorption is used to attenuate low- and high-energy components of the primary beam.

#### 3.1.3. Converging Optics with Bent HOPG/HAPG Crystals

The utilization of HOPG/HAPG crystals as bent monochromators for X-ray focusing has been explored in various studies and have ranged from cylindrical, conical, paraboloid, and doubly bent toroidally shaped to ellipsoidally bent crystals [33,38,39,40,41,42,43,44]. In addition to HOPG, thin films of HAPG, exhibiting even lower mosaicity, have been shown to be a promising material for focusing optics due to their ability to be strongly bent into small radii and virtually any shape [32,37,38]. The conventional procedures reported to produce HOPG involve annealing of pyrolytic graphite under pressure on convex/concave graphite pistons at temperatures around 3000 °C [32,42,91]. Herein, pyrolytic graphite is obtained on a heated substrate from the pyrolysis of hydrocarbons with carbon deposition from the gaseous phase [32,42,91]. Through changes in the parameters of annealing under pressure, highly flexible thin HOPG films can be obtained that are stripped of their bulk [32,42,91]. These films can easily be stacked and mounted on complex-shaped substrates. Significant upgrades to the annealing process results in HAPG monofilms exhibiting mosaicity of about 0.1° [32]. These films can be plastically bent at room temperature without significant structural changes or residual stress from deformation [32]. The construction of substrates into molds of desired shapes involves various approaches depending on the material type: e.g., computer-controlled precision machines are generally utilized for metal-based substrates such as aluminum [32].

In our work, the X-ray focusing optics are composed of mosaic graphite crystal layers (HOPG/HAPG parameters, as specified by Optigraph GmbH, Berlin, Germany, and Grigorieva et al. [32]) deposited on a concave ellipsoidal substrate. The ellipsoidal arrangement is rotationally symmetric about the primary beam axis. The concave inner region is employed for the Bragg reflection process from the surface and bulk of the crystal. Figure 8b,c display the simple schematic and cross-sectional view of the optics setup. The graphite crystal layer (with parameters mentioned in Table 3) is placed on an aluminum ellipsoidal substrate with a major curvature of $b_{0}=500$ mm and a minor curvature of $a_{0}=10$ mm. The substrate assembly is placed inside an aluminum shell since the practical manufacturing process necessitates constructing the ellipsoidal optics design in two halves. The deposition of the crystal layer is carried out on each half using a former that has a negative shape mirroring the required ellipsoidal geometry [41]. One of the previous studies employing ellipsoidal optic shapes utilized an ultra-high-speed diamond-turning machine to construct an aluminum former with a surface roughness of a few nanometers and an ellipsoidal inner cavity using a classical inside turning method [41].

In our work, this assembly is enclosed inside an external optics shield composed of lead (Pb). A beam-stop made up of Pb and with dimensions shown in Figure 8c is used to block the direct passage of X-rays from the optical system. Two external shielding elements, composed primarily of Pb, secure the system, and the beam-stop is mounted using aluminum supports. The two external elements further serve as shields against the direct beam effects and also against other scattered events. This structure also collimates the primary beam, limiting the passage of X-rays to those necessary for interacting with the crystal layers.

The source position is selected to converge X-rays downstream at the focal length (crystal-center to target distance of f=320 mm) based on the design parameters of the optics, with curvatures $b_{0}$ and $a_{0}$, additional space requirements to accommodate other imaging components (i.e., detectors, apertures, etc.), and sufficient working distance for experimentation. With rotational symmetry of the ellipsoid optics about the major axis, the relation between *f* and the incidence angle on the crystal surface, $\theta_{inc}$, can be expressed as
(5)$$\theta_{inc}=\arccos\left(-\frac{b_{0}\sin\phi -f}{\sqrt{(a_{0}^{2}\cos^{2}\phi +{\left(b_{0}\sin\phi -f\right)}^{2}}}\right),$$
where ϕ∈[0,2π] is determined by the photon incidence position on the crystal surface. For the given geometry, the average incident photon angle and its spread on the crystal surface are estimated (using MC simulations) as $\theta_{inc}=1.78{}^{\circ}\pm 0.1{}^{\circ}$. This value is close to $\theta_{B}=1.79{}^{\circ}$ in the (002) GrO reflection for $K_{\alpha }$ tungsten characteristic X-rays, and its spread is within the lowest simulated mosaicity, i.e., m=0.12°. Considering this, the required dimensions of the beam-stop and the length of the crystal surface are optimized. Consequently, the external and central diameters of the beam-stop and the crystal length are determined as 14.8 mm, 17.8 mm, and 40 mm, respectively.

Within the focusing configuration used here, the primary diffraction process is through Bragg reflection geometry. The minimal presence of Laue-diffracted X-rays is suppressed further due to the beam-stop geometry and shielding elements implemented here. In such a geometric configuration utilizing HOPG/HAPG crystals, well-collimated incident and reflected beams are expected to enhance contributions from the (002)-reflection. Due to varying sets of lattice planes presenting different interplanar distances, the Bragg angles for the same X-ray wavelength differ. A well-collimated geometry and large focal length in a one-to-one configuration (equal source-to-crystal and crystal-to-target distances) would suppress other reflexes (e.g., in 004-reflection, $\theta_{B}=3.5{}^{\circ}$ for tungsten $K_{\alpha }$ X-rays). Conversely, a lack of beam collimation, substantial misalignments, and an increasing mosaic spread may result in the broadening of the focused beam by permitting a wide range of incident and scattering angles. This is due to the Bragg diffraction becoming increasingly diffuse from multiple reflexes as the Bragg condition is fulfilled, along with mosaic effects [34,87]. The (002)-lattice planes in these crystals inherently exhibit the strongest diffraction (highest reflection intensity) compared to the other planes, and crystallites tend to preferentially orient parallelly along the film deposition layers with a small spread, particularly under low mosaicity conditions [40,41,43,44,87,88,89]. However, the impact of deviating effects, such as deviations from the assumed focusing conditions or imperfections in the produced crystal quality, will need evaluation through future experiments, including quality characterization tests.

Photon tracking and interactions within the simulation framework are illustrated in Figure 8a, which shows the Bragg-reflected X-rays converging downstream of the optical system along with other EM physical processes occurring. The results of the ellipsoidal focusing performance with different mosaic spreads are presented in Figure 1, Figure 2b, and Table 2.

#### 3.1.4. Rocking Curve Analysis

The performance of the Bragg-reflection simulation model is evaluated using rocking curve analysis performed on planar GrO. Simulations are performed using monochromatic photon beams with energies of 59 keV, corresponding to a Bragg angle of $\theta_{B}=1.79{}^{\circ}$, and, additionally, with 8.03 keV beams, corresponding to a Bragg angle of $\theta_{B}=13.29{}^{\circ}$ [32]. Three mosaic spread parameter values are used, i.e., m=0.12°, 0.24°, and 0.46°. The Bragg reflection of the 59 keV beam is analyzed with each planar crystal layer placed on a 5 mm thick aluminum substrate, whereas for the 8.03 keV beam, 5 mm thick glass substrates are utilized. Intensities of the incident and reflected photons are recorded in histograms for 51 angular stepping positions around the Bragg angle with an interval of 0.05°. Using this, the FWHM of the rocking curve δ, peak reflectivity $R_{\mathrm{peak}}$, and integrated reflectivity $R_{i}$ are determined.

Following Equations (Equation 2) and (Equation 4), the theoretical peak reflectivity is calculated at the ideal Bragg angle (Δ=0) as per the reflectivity formula [84]:(6)$$r\left(\Delta =0\right)=a_{p}/\left(1+a_{p}+\sqrt{1+2a_{p}}\coth\left(b_{p}\sqrt{1+2a_{p}}\right)\right),$$
(7)$$a_{p}=W\left(\Delta =0;m\right)Q/\mu_{tot},$$
(8)$$b_{p}=T\mu_{tot}/\sin\theta_{B},$$
where $\mu_{tot}$ is the total attenuation coefficient from photoelectric absorption and attenuation due to Compton scattering, and *T* is the crystal thickness. These results are shown in Figure 2a and Table 1.

#### 3.1.5. Tumor-Bearing Mouse Model

For the in silico assessment of the proposed X-ray optics intended for pre-clinical XFI applications, a voxelized 3D model of tumor-bearing mice is implemented, referring to a methodology similar to [17]. The voxel model is based on a whole-body segmented mouse model (Digimouse, 0.1 mm cubical voxels, 380×992×208 matrix size) generated through coregistered CT and cryosection image data of a 28 g nude normal male mouse [59,60]. The dimensions of the voxel model are reduced to a matrix size of 190×496×104 to optimize memory utilization, thereby resulting in 0.2 mm cubical voxels. The material properties and composition of the model are determined based on the Geant4 database. The mouse is embedded inside a Plexiglas mount (mouse restrainer) of 1 mm thickness and mounted on a rotation stage arrangement composed primarily of aluminum. Figure 9a shows an example of the simulation setup, where the focused X-rays from the ellipsoidal GrO interact with the voxelized mouse model. Here, only the central sagittal slice is visualized. Figure 9b shows the volume rendering of the implemented Digimouse model with segmented organs.

#### 3.1.6. Detector Model

Two detector models, i.e., corresponding to the silicon drift detector (SDD) and cadmium telluride (CdTe) detector, were built using derived sensitive detector classes from the Geant4 base class *G4VSensitiveDetector*. For XFI simulations involving multiplexed (multi-tracking) Pd and Ba contrast agents, six SDDs (XR-100FastSDD, Amptek Inc., Bedford, MA, USA) were used, each comprising a 1 mm thick silicon sensor and a 70 ${\mathrm{mm}}^{2}$ detector area—collimated to a 40 ${\mathrm{mm}}^{2}$ active area [78]. The energy resolution $\left(\sigma_{det}\right)$ of Amptek SDDs typically ranges from 230 eV to 300 eV FWHM at Pd $K_{\alpha }$ energies and from 280 eV to 350 eV FWHM for Ba $K_{\alpha }$ energies depending on the maximum output count rate (which is dependent on the processor peaking time) and sensor temperature [78,92]. Simulations with Fe55 radioactive sources were conducted (refer to Appendix A) to compare the implemented SDD detector model with the reference [78,92]. The energy resolution ($\Delta E_{\mathrm{fit}}$, in units of eV FWHM) at 5.9 keV $K_{\alpha }$ photopeak (arising from decay to Mn55) is calculated from the simulated spectrum for different levels of simulated electronic noise. As seen in Appendix A, the $\Delta E_{\mathrm{fit}}$ estimates fall within the expected range of typical resolution values (∼124 eV FWHM to ∼148 eV FWHM, depending on the output count rate) reported in the references [78,92]. Additionally, the simulated spectra align well with the measured spectra reported in [78,92]. For XFI simulations, a conservative value of the electronic noise (equivalent noise charge, ENC=75 eV) is assumed instead of the typical 20–50 eV. It is noteworthy to mention that opting for ENC=75 eV, as opposed to 24 eV, results in an approximately 9% decrease in the significance (Z, see Equation (Equation 9)) of Pd $K_{\alpha }$ fluorescence in our simulations.

For singleplex XFI simulations with Ba contrast agents, the utility of the higher-stopping-power CdTe sensors (compared to silicon sensors) is explored [93,94]. The six CdTe detectors (XR-100CdTe, Amptek Inc., Bedford, MA, USA) each comprised a 1 mm thick CdTe sensor with a 25 ${\mathrm{mm}}^{2}$ active area [79]. The energy resolution $\left(\sigma_{det}\right)$ of the CdTe detector around Ba $K_{\alpha }$ energies is typically on the order of 450 eV FWHM, with some detectors reaching up to 700 eV FWHM (typically ≤800 eV FWHM at 59.5 keV) [79,93,94]. Simulations with Co57 radioactive sources were conducted (refer to Appendix A) to compare the implemented CdTe detector model with the measured data provided by Amptek. Sensitivity analysis is used to assess the sensitivity of the implemented detector model to variations in detector-specific effects and geometric components. The implemented CdTe model exhibits good agreement with the Amptek data, as seen in Appendix A.

### 3.2. Procedure for In Silico XFI Analysis

#### 3.2.1. Imaging Modes

Two imaging modes are used, i.e., a *low-dose* and a *high-dose* mode. These modes are intrinsically determined by the photon flux density per scan position, scanning area per projection, and the number of XRF projections per tomography scan. These factors, driven by the efficiency of the proposed optics, determine the total allowable time that can be allocated for a single XRF tomography scan. A photon flux density of $9.7\times 10^{7}$ $\mathrm{photons}\cdot {\mathrm{mm}}^{-2}\cdot s^{-1}$ is achieved in the 600 μm (transversal FWHM) diameter of the focused beam with m=0.12°. In the *low-dose* mode, $2.7\times 10^{7}$ photons are used per step position, while within the *high-dose* mode, $2.7\times 10^{8}$ photons are used per step position.

#### 3.2.2. Tumor Lesion Targets

The tumor lesion targets are implemented at varying agent concentrations in simulations and comprise the Geant4 soft tissue material (*G4_TISSUE_SOFT_ICRP*). The diameters of the spherical tumor targets are 0.5 mm, 1 mm, 1.25 mm, 2 mm, 5 mm, and 10 mm. Three organ regions are selected for focused beam scanning, i.e., the subcutaneous tissue (at the lower right flank), liver (i.e., central/upper abdominal region), and left kidney (i.e., the abdominal cavity region). For tumor lesions corresponding to larger target sizes protruding through the subcutaneous tissue, an additional 0.5 mm thick layer of skin comprised of adipose tissue (*G4_ADIPOSE_TISSUE_ICRP*) is added [17]. The mouse model is positioned, simulating a translation scan, in a way that the tumor target is always at the scan center of the system. The inset image in Figure 9a highlights a 1.25 mm tumor lesion (see the golden sphere indicated by the red arrow) embedded inside the liver.

#### 3.2.3. Contrast Agents and Concentrations

The simulations involving SDDs included a multiplex panel consisting of palladium (Pd) and barium (Ba) contrast agents, while simulations with CdTe detectors included only Ba as the contrast agent. For *low-dose* imaging mode, dilutions of Pd and Ba in the tissue targets include concentrations of 0.01 wt%, 0.033 wt%, 0.05 wt%, 0.1 wt%, 0.33 wt%, 0.66 wt%, and 1 wt%. Additional concentrations of 0.0066 wt% and 0.0033 wt% are used for *high-dose* imaging.

#### 3.2.4. Detector Arrangement and Positioning

For each of the simulations with SDDs and CdTe detectors, six detectors are utilized and are positioned upstream of the focused beam propagation. The azimuth angles for SDDs and CdTe detectors are set to 140° and 150°, respectively. Detector positioning at these angles provides adequate clearance for mouse rotational and translational scanning while maintaining the shorter detector-to-target distances necessary for large solid angle coverage. Specifically, for CdTe detectors, the intrinsic fluorescence photons and escape events substantially contribute to the Ba $K_{\alpha }$ fluorescence energy region. Therefore, to minimize the contribution of multiple Compton-scattered photons, intrinsic CdTe fluorescence, and escape events, a 150° angle was determined as optimal. The detectors are mounted on a holder arrangement with the polar angle of the detectors in the upper and lower rows set to 30° and a detector-to-target distance of 32 mm. The detectors in the central row are at a detector-to-target distance of 26 mm to provide clearance for the translation and rotation of the mouse during the XFI tomography scan. The detector arrangement (side-view) is shown in Figure 9a.

#### 3.2.5. Radiation Dose

Dose scoring was implemented by creating dose accumulators for each tissue type using the *G4Accumulable* class of the Geant4 toolkit, which subsequently yielded the corresponding doses to tissues and organs in the mice model. The organ masses were determined based on the number of voxels, voxel volume, and material density associated with each tissue type of the segmented model. The total dose to tissues and organs was thereby calculated from the energy deposited per hit position stored in the accumulators. The organ doses per scan position were compared between the *low-dose* and *high-dose* XFI modes for the three organ regions (i.e., subcutaneous, liver, and kidney), as shown in Figure 7.

### 3.3. Statistical Analysis

In order to quantify the significance of the XRF signals corresponding to the simulated spectral data, one-tailed hypothesis tests are performed. Herein, the null hypothesis ($H_{0}$) posits that the presence of any XRF-like signal features in the background arises solely through fluctuations in the background processes and not from the fluorescence processes within the contrast agents [10,15,17,20]. The *p*-value thereby gives the probability that the observed effect is only a fluctuation in the background processes and not fluorescence, assuming $H_{0}$ is true. For the sake of handling small *p*-values conveniently, one can alternatively estimate the statistical significance (Z, which relies on *p*-values) within the expected signal range to assess the detectability of fluorescence signal over the background [10,17,20]. The significance (Z), expressed in units of one standard deviation of background, can be approximated as [10,15]
(9)$$Z\approx \frac{N_{F}}{\sqrt{N_{B}}},$$
where $N_{F}$ and $N_{B}$ represent the estimated number of XRF and background photons, respectively, within an energy range centered around the corresponding energies of the fluorescence lines and with a width dictated by the detector’s energy resolution. In this work, two thresholds are utilized for significance testing of the XRF detectability, i.e., Z=5, corresponding to a false positive probability of $p=2.868\times 10^{-5}\%$, and Z=3, corresponding to a false positive probability of p=0.135%. To determine $N_{F}$ and $N_{B}$, peak fitting is performed using fitting models based on the ratio of fluorescence cross-sections from the *xraylib* database [10,95]. The total fluorescence cross-section for the relevant transition lines of the contrast agents was calculated through a weighted integration utilizing the energy probability distribution of the incident polychromatic X-rays. For spectra fitting, a model consisting of a sum of Gaussians centered around the energy of the respective fluorescence lines is used and weighted by the product of the cross-section of the transition lines and the detector’s efficiency [10]. The width of each Gaussian is defined by the detector’s energy resolution $\sigma_{det}$, where the fitting energy range is at least $\pm 3\sigma_{det}$ around the respective fluorescence peak. The fit model also incorporates a third-order polynomial to account for the contributions from the continuous Compton background. The results for fluorescence significance (Z) with SDDs and CdTe detectors in the *low-dose* imaging mode are presented in Figure 3 and Figure 5, respectively. The Z values with SDDs and CdTe detectors in the *high-dose* mode are presented in Figure 4 and Figure 6, respectively.

## 4. Conclusions

This study employs Monte Carlo simulations to investigate the feasibility of focusing hard X-rays utilizing mosaic graphite crystals for high-resolution XFI applications under in vivo conform conditions. Quantitative predictions for small-animal imaging scenarios demonstrate the potential for sub-millimeter-level resolution with high sensitivity and low radiation doses, suggesting promising advancements in benchtop systems. Efficient hard X-ray lenses may enable high-resolution XFI with a broader range of heavier marker substances (42≤Z≤64) using conventional or microfocus solid-target X-ray tubes. However, further research and experimentation are necessary to evaluate and validate this concept fully. This is particularly crucial considering the challenges in accurately predicting the efficiency of such crystals for real-case experiments through simulations alone.

Future studies focusing on X-ray optics for XFI or other imaging approaches might also benefit from the findings and simulation methodology presented in this preliminary investigation. Building upon current findings, our future work will employ further simulations and experimentation to optimize X-ray optics design, with the aim of improving performance and realizing nested lens systems. Furthermore, we aim to broaden the scope by utilizing this concept for large-sized objects.

## Figures and Tables

**Figure 1 ijms-25-04733-f001:**
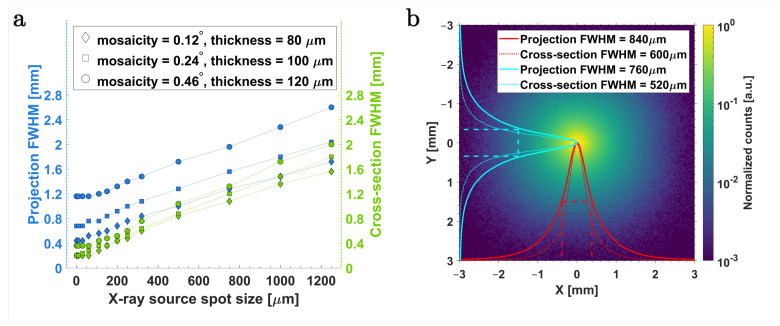
Spatial properties of the proposed ellipsoidal optics. (**a**) The dependence of the projection (blue) and cross-sectional (green) FWHM of the focused beam on the size (FWHM) of the X-ray source spot. (**b**) Normalized intensity distribution on the focal plane for source size of 320 μm FWHM and m=0.12°. The red and cyan solid/dotted lines represent the focused beam’s horizontal and vertical projection/cross-section FWHM.

**Figure 2 ijms-25-04733-f002:**
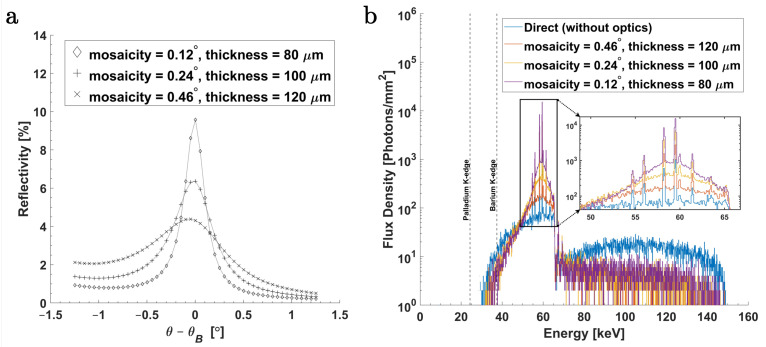
Rocking curve and spectral characterization of the Bragg-reflecting GrO. (**a**) Rocking curve evaluation on planar GrO considering a source size of 320 μm FWHM and 59 keV monochromatic beams. (**b**) Comparison of the photon flux density in units of photons per ${\mathrm{mm}}^{2}$ and at f=320 mm between the direct X-rays (blue) and the focused X-rays using ellipsoidal GrO.

**Figure 3 ijms-25-04733-f003:**
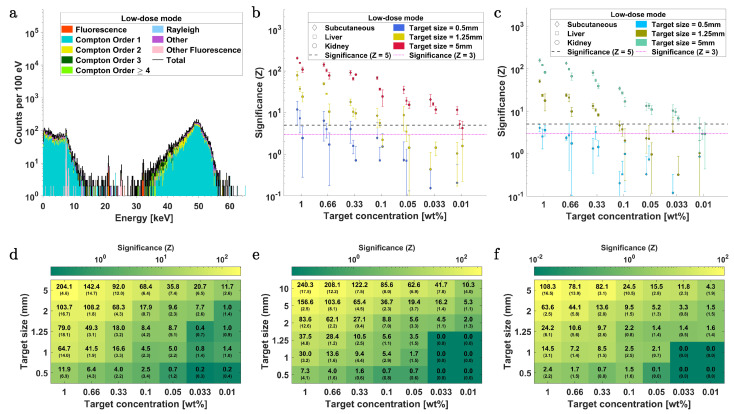
Comparison of significance (Z, see Equation (Equation 9)) at *low-dose* mode XFI across varying tumor lesion (target sphere) sizes within the subcutaneous, liver, and kidney regions. (**a**) A typical simulated XRF spectrum per scan position corresponding to the voxelized 3D mouse model and featuring palladium (Pd) and barium (Ba) as contrast agents within the target spheres. Significance (mean values ± standard deviation) of (**b**) ${\mathrm{Pd-K}}_{\alpha }$ and (**c**) ${\mathrm{Ba-K}}_{\alpha }$ fluorescence at varying agent concentrations within tumor lesions of 0.5 mm, 1.25 mm, and 5 mm diameter positioned inside the three organ regions. The black dashed line indicates a significance (Z) cutoff of Z=5, and the magenta dotted line indicates Z=3. Significance (mean values and standard deviation in brackets) of ${\mathrm{Pd-K}}_{\alpha }$ fluorescence for varying agent concentrations and target sizes within the (**d**) subcutaneous, (**e**) liver, and (**f**) kidney regions.

**Figure 4 ijms-25-04733-f004:**
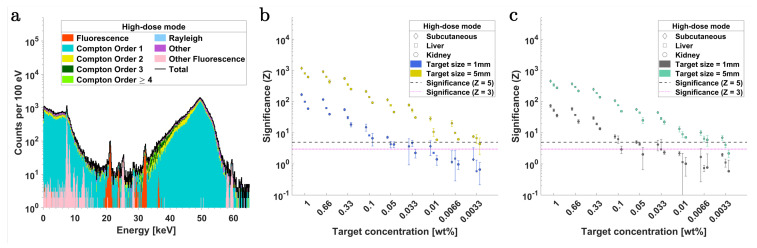
Fluorescence imaging of palladium (${\mathrm{Pd-K}}_{\alpha }$) and barium (${\mathrm{Ba-K}}_{\alpha }$) contrast agents at *high-dose* mode with SDDs. (**a**) A typical simulated XRF spectrum per scan position. Significance (mean values ± standard deviation) of (**b**) ${\mathrm{Pd-K}}_{\alpha }$ and (**c**) ${\mathrm{Ba-K}}_{\alpha }$ fluorescence at varying agent concentrations within tumor lesions (1 mm and 5 mm diameter) positioned inside three organ regions.

**Figure 5 ijms-25-04733-f005:**
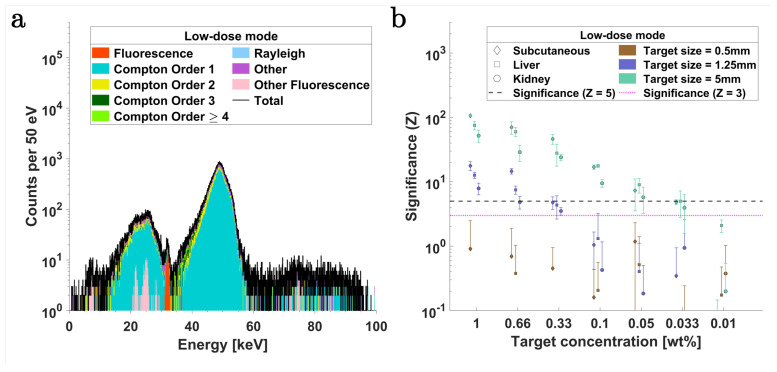
Fluorescence imaging of barium (${\mathrm{Ba-K}}_{\alpha }$) contrast agents at *low-dose* mode with CdTe detectors. (**a**) A typical simulated XRF spectrum per scan position. (**b**) Comparison of ${\mathrm{Ba-K}}_{\alpha }$ significance (mean values ± standard deviation) for varying agent concentrations and tumor sizes (0.5 mm, 1.25 mm, and 5 mm diameter) within the subcutaneous, liver, and kidney regions.

**Figure 6 ijms-25-04733-f006:**
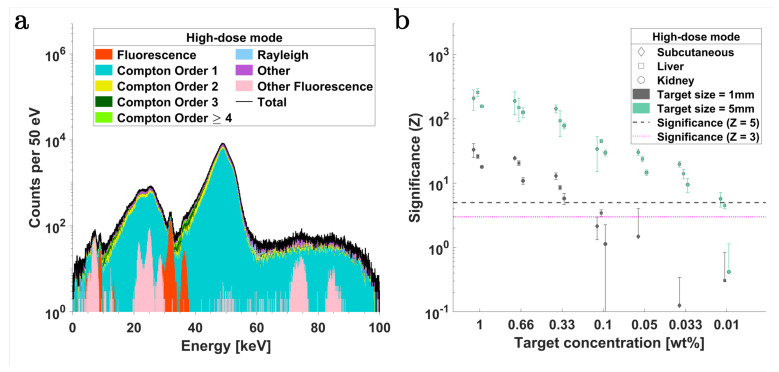
Fluorescence imaging of barium (${\mathrm{Ba-K}}_{\alpha }$) contrast agents at *high-dose* mode with CdTe detectors. (**a**) A typical simulated XRF spectrum per scan position. (**b**) Significance (mean values ± standard deviation) of ${\mathrm{Ba-K}}_{\alpha }$ fluorescence at varying agent concentrations within tumor lesions (1 mm and 5 mm diameter) positioned inside three organ regions.

**Figure 7 ijms-25-04733-f007:**
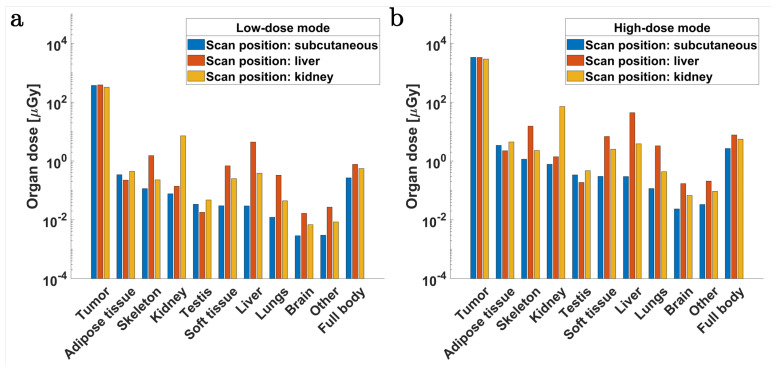
Comparison of organ doses across *low-dose* and *high-dose* mode fluorescence imaging. (**a**) Organ doses in the *low-dose* mode and (**b**) in the *high-dose* mode are shown for the three organ scan positions that include the tumor target, i.e., (blue) subcutaneous, (orange) liver, and (golden) kidney region. Herein, the tumor dose corresponds to a lesion size of 1 mm in diameter.

**Figure 8 ijms-25-04733-f008:**
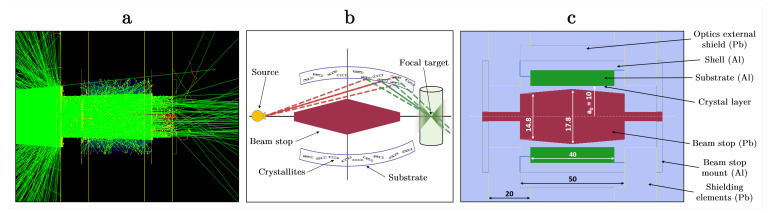
(**a**) Illustration depicting photon tracking and interactions with the ellipsoidal GrO setup in Geant4. A total of $10^{4}$ simulated photon histories are shown, and Bragg-reflected photons can be seen converging downstream of the optics setup. Trajectories of other scattered events can also be seen. (**b**) A simplified illustration of the optical setup. (**c**) Schematic diagram (not to scale) showing a cross-sectional view of the optics setup. All units are in mm.

**Figure 9 ijms-25-04733-f009:**
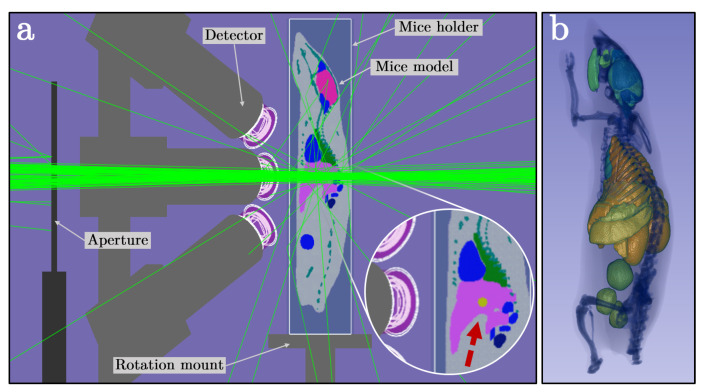
(**a**) An exemplary image of the simulation setup. The red arrow (see the inset image) highlights the tumor lesion (golden sphere) embedded within the mouse liver. (**b**) Volume rendering of the whole-body voxelized mouse model (Digimouse [59,60]) used for the MC simulations.

**Table 1 ijms-25-04733-t001:** Comparison of GrO parameters between reference data values (taken from Grigorieva et al. [32]), theoretical values, and those evaluated through MC simulations. The simulated results correspond to rocking curve evaluations with a 59 keV ($\theta_{B}=1.79{}^{\circ}$) and an 8.03 keV ($\theta_{B}=13.29{}^{\circ}$) monochromatic photon beam. Theoretical peak reflectivity $R_{\mathrm{peak}}^{\mathrm{theo}}$ is calculated as per Equation (Equation 6).

*m* [°]	Thickness [μm]	$\delta^{\mathrm{MC}}$ [°]		$R_{i}^{\mathrm{MC}}/R_{\mathrm{peak}}^{\mathrm{MC}}$ [mrad/%]		$R_{\mathrm{peak}}^{\mathrm{theo}}$ [%]		δ [32] [°]	$R_{i}$ [32]/$R_{\mathrm{peak}}$ [32] [mrad/%]	Crystal [32]
		59 keV	8.03 keV	59 keV	8.03 keV	59 keV	8.03 keV	8.03 keV	8.03 keV	
0.12	80	0.3	0.6	0.5/9.5	2.2/21	16.5	46.5	0.6	0.85/8.2	HOPG (Thin films)
0.24	100	0.6	0.8	0.7/6.4	2.8/20	10.8	34.7	0.8	1/6.95	HOPG (Thin films)
0.46	120	1.25	1.2	1/4.4	3.4/16.2	6	24.4	0.9	1.1/6.8	HOPG (Thin films)

δ: FWHM of rocking curve, $R_{i}$: integrated reflectivity, and $R_{\mathrm{peak}}$: peak reflectivity.

**Table 2 ijms-25-04733-t002:** Performance of the ellipsoidal GrO. The X-ray focusing properties and spectral broadening at f=320 mm are compared for the three mosaicities considering a source size of 320 μm FWHM. The results correspond to the Monte Carlo (MC) simulations performed on Geant4 and ray-tracing (RT) simulations on *mmpxrt*.

*m* [°]	Projection/Cross-Section ${\mathrm{FWHM}}^{\mathrm{MC}}$ [mm]	${\mathrm{Gain}}^{\mathrm{MC}}$ [#]	$\Delta E_{5\%}^{\mathrm{MC}}$ [keV]	$\Delta E_{5\%}^{\mathrm{RT}}$ [keV]	$\Delta E_{5\%}^{\mathrm{MC}}/E_{\mathrm{peak}}^{\mathrm{MC}}$ [%]	$\Delta E_{5\%}^{\mathrm{RT}}/E_{\mathrm{peak}}^{\mathrm{RT}}$ [%]
0.12	0.84/0.6	14	5.8	8.9	9.7	15
0.24	1.1/0.64	6.5	7.1	10.6	11.9	17.9
0.46	1.5/0.8	2.6	9.4	11	15.8	18.4
Direct X-rays	-	1	16.5	-	27.7	-

$\Delta E_{5\%}$: Energy bandwidth (full-width at 5% or 1/20 of maximum) of the focused beam. $\Delta E_{5\%}/E_{\mathrm{peak}}$: Ratio of the energy bandwidth ($\Delta E_{5\%}$) to the peak energy of the focused beam.

**Table 3 ijms-25-04733-t003:** Summary of the primary optical parameters employed within the MC simulations. The ellipsoidal GrO is applied to focus X-rays in the XFI simulations, while planar GrO is used for the rocking curve analysis.

Parameter	Value	
	Ellipsoidal GrO	Planar GrO
Mosaic spread *m* [°]	0.12, 0.24, 0.46	0.12, 0.24, 0.46
Crystal thickness *T* [μm]	80, 100, 120	80, 100, 120
Peak energy $E_{peak}$ [keV]	59	59/8.03
Bragg angle $\theta_{B}$ [°]	1.79	1.79/13.29
Diffraction order *n*	1	1
Miller indices hkl	002	002
Lattice spacing $d_{hkl}$ [nm]	0.3354	0.3354

## Data Availability

The raw data supporting the findings of this study will be made available by the authors without any undue reservations. All relevant data supporting the findings of this study can be found within the article or Appendix A.

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
