# Peer review of "High-Spatial-Resolution Benchtop X-ray Fluorescence Imaging through Bragg-Diffraction-Based Focusing with Bent Mosaic Graphite Crystals: A Simulation Study"

_ijms, 2024, doi:10.3390/ijms25094733_

Round 1
Reviewer 1 Report
Comments and Suggestions for Authors
The article is devoted to a current topic - modeling of desktop X-ray fluorescence imaging of high spatial resolution using the Bragg diffraction focusing method with curved mosaic graphite crystals. The article corresponds to the journal's profile.
In my opinion, several points require clarification:
1. Why, when modeling the installation, did the authors choose a tube with an anode and cathode made of tungsten? Is this due only to the refractoriness of tungsten? Tungsten is a rather exotic material for anada, although the cathode of all tubes is tungsten.
2. Why did the modeling take into account only the (002) reflection for graphite? Would observing multiple Bragg reflexes increase or decrease resolution?
3. How should the graphite crystal needed for this installation be produced in practice? Like a curved monochromator?
Reviewer 2 Report
Comments and Suggestions for Authors
The paper presents a Monte Carlo simulation of the X-ray fluorescence using an ellipsoidal lens system composed of mosaic-graphite crystals. Specifically, the author investigated the x-ray focusing capability of Bragg-reflecting crystals with three different mosaic spreads and corresponding crystal thicknesses. Then simulations were performed for digital mouse models to evaluate the significance values using palladium (Pd) and barium (Ba) nanoparticles as contrasting agents under high and low doses. Two detectors were used including the silicon drift detector and cadmium telluride detectors. Based on the results, the designed optical setup can detect a minimum total mass of Pd nanoparticles of a few hundred nanograms. The simulations can provide important design principles for improving high-resolution in vivo x-ray fluorescence imaging with mid-Z elements. The paper is well-written, and the results are clearly presented. I have some suggestions in terms of the arrangement of the content.
The authors mostly put the theoretical part of the simulation in the last section 3. Materials and Methods. The manuscript can be more approachable if the authors can move some of the important equations and definitions upfront. For example,
1. The author can combine Figure 8 with Figure 1 to show the focusing optics at the beginning of the paper. As the author mentioned in the paper, one of the novelty of the paper is using mosaic graphite optics for focusing X-rays. Putting Figure 8 in the front can give the audience a clear picture of the set-up geometry.
2. I would also encourage the authors to add the important equations in section 3.1.1 about the Monte Carlo simulations to the earlier sections where the calculation results are presented.
3. Similarly, for section 3.3, the definition of the significance value is not mentioned when the results are presented.
